# Optimization of Callus Induction and Shoot Regeneration from Tomato Cotyledon Explants

**DOI:** 10.3390/plants12162942

**Published:** 2023-08-14

**Authors:** Olha Yaroshko, Taras Pasternak, Eduardo Larriba, José Manuel Pérez-Pérez

**Affiliations:** Instituto de Bioingeniería, Universidad Miguel Hernández, 03202 Elche, Spain; oyaroshko@umh.es (O.Y.);

**Keywords:** in vitro culture, de novo shoot formation, cytokinin, auxin, *Solanum lycopersicum* L.

## Abstract

Cultivated tomato (*Solanum lycopersicum* L.) is one of the most important horticultural crops in the world. The optimization of culture media for callus formation and tissue regeneration of different tomato genotypes presents numerous biotechnological applications. In this work, we have analyzed the effect of different concentrations of zeatin and indole-3-acetic acid on the regeneration of cotyledon explants in tomato cultivars M82 and Micro-Tom. We evaluated regeneration parameters such as the percentage of callus formation and the area of callus formed, as well as the initiation percentage and the number of adventitious shoots. The best hormone combination produced shoot-like structures after 2–3 weeks. We observed the formation of leaf primordia from these structures after about 3–4 weeks. Upon transferring the regenerating micro-stems to a defined growth medium, it was possible to obtain whole plantlets between 4 and 6 weeks. This hormone combination was applied to other genotypes of *S. lycopersicum*, including commercial varieties and ancestral tomato varieties. Our method is suitable for obtaining many plantlets of different tomato genotypes from cotyledon explants in a very short time, with direct applications for plant transformation, use of gene editing techniques, and vegetative propagation of elite cultivars.

## 1. Introduction

Tomato (*Solanum lycopersicum* L.) is the most widely grown vegetable in the world. According to the latest FAOSTAT report [1], world tomato production exceeded 189 million tons in 2021, with China, India, and Turkey being the largest producers. Global demand for tomatoes is expected to increase in the future. However, in many of the current tomato growing areas, the consequences of human-induced climate change are expected to negatively affect crop productivity due to an increase in average temperature and irregular rainfall in these areas [2]. Therefore, the development of new tomato cultivars with greater tolerance to abiotic (i.e., drought, temperature) and biotic stresses is a priority breeding goal that can be accelerated using new genome editing strategies (i.e., CRISPR/Cas-based approaches) [3]. Successful implementation of these strategies requires the regeneration of whole plantlets from tissue explants, a process that generally relies on hormonal induction of de novo shoot formation from tissue explants. The efficiency of regeneration depends on several factors, including the type of explant, the culture conditions and composition of the regeneration medium, and the genotype and physiological state of the mother plant used [4].

In recent years, the dwarf tomato cultivar Micro-Tom (MT) has been established as a model for functional genomics research due to its small size and short life cycle [5], which allows it to be grown indoors at high densities, and the implementation of high-throughput genetic approaches such as publicly available mutant collections, efficient *Agrobacterium tumefaciens*-mediated transformation protocols, mapping-by-sequencing, and precise genome editing [6,7]. Other classic tomato cultivars that have been widely used in research are M82 and Moneymaker (MM). These three cultivars concentrate most of the genetic resources available in tomatoes, facilitating their use as experimental models [3]. However, despite the remarkable morphological differences among these three cultivars due to background-specific developmental mutations, genetic diversity in modern cultivated tomatoes is limited. In a recent study, Mata-Nicolás et al. (2020) characterized a collection of diverse *S. pimpinellifolium*, *S. lycopersicum* var. *cerasiforme*, and *S. lycopersicum* var. *lycopersicum* that represent the genetic and morphological variability of tomato at its centers of origin and domestication [8].

Shoot regeneration of various tomato cultivars and wild tomato species by indirect organogenesis (i.e., callus) has been widely reported, with contrasting results due to strong genotype dependence [9]. Indeed, it was recently found that enhanced in vitro shoot regeneration of the wild relative *S. pennellii* over commercial cultivars such as MT depends on three genomic regions of *S. pennellii*, one of which is associated with enhanced expression of the shoot-related genes *WUSCHEL* (*WUS*) and *SHOOT MERISTEMLESS* (*STM*) [10]. In recent years, several reports have been published on the implementation of cost-effective and reproducible *A. tumefaciens*-mediated transformation systems for different commercial tomato cultivars (M82, MT, Rio Grande, Pusa Ruby, Arka Vikas, etc.), all based on efficient regeneration protocols with different proportions of auxin and cytokinin (CK) in the medium depending on the cultivar used [11,12,13,14,15]. Only a few studies have focused on the systematic optimization of several tissue culture parameters to establish a more efficient and reproducible shoot-regeneration system in tomatoes [16,17].

In this study, we evaluated the effectiveness of several tissue culture parameters for de novo shoot regeneration in different tomato cultivars, including modern tomato cultivars and heirloom tomato genotypes from the centers of origin (Perú) and domestication (México) of tomatoes. We found remarkable differences in both callus formation and de novo shoot induction, which depended on both the position of the explants and the genotype of the mother plant. In addition, we performed a detailed histology characterization of de novo shoot formation in M82 cotyledon explants as an example of a tomato genotype with a high in vitro regeneration ability.

## 2. Results

### 2.1. Optimization of Callus and Shoot Formation in M82 and MT Cotyledon Explants

We found that the area of tomato cotyledon explants increased approximately three-fold during the experiment in indole-3-acetic acid (IAA)-dependent manner (*p*-value = 0.000; multifactorial ANOVA [MANOVA]; *n* = 259; Appendix A), but with a non-significant effect of zeatin (ZT) levels (*p*-value = 0.453) and/or genotype (*p*-value = 0.207). Callus formation was dependent on both IAA and ZT levels (*p*-value = 0.000; MANOVA; *n* = 584), with a small effect of genotype (*p*-value = 0.071). In MT, callus formation in cotyledon explants incubated without IAA was observed in a lower proportion of explants than in M82, except at high ZT (3 mg/L) (Figure 1a). On the other hand, callus growth, measured as callus area at the end of the experiment, was slightly dependent on exogenous ZT levels (*p*-value = 0.031; MANOVA; *n* = 186) and on the interaction between ZT and IAA (*p*-value = 0.042) (Figure 1b). Indeed, high ZT levels (3 mg/L) produced significantly smaller calluses in both genotypes (*p*-value = 0.001). We found a positive and significant correlation between callus area and explant growth (r = 0.593; *p*-value = 0.000) (Figure 1c), which is consistent with the increase in callus area accounting for most of the explant growth. Calluses were mainly formed at the cut ends of the explants. In this region, the callus tissue formed had a creamy yellow color and a compact structure protruding from both abaxial and adaxial sides of the explant (Figure 1d). Next, additional tissue outgrowth was observed near the cotyledon margin and the midrib, which later extended into the whole lamina (Figure 1e). Based on the external appearance of the tissue, we distinguished between soft callus and dense callus, the former being friable in nature and separable by light touch with tweezers.

Interestingly, we found striking differences in de novo shoot formation from these calluses depending on genotype and exogenous ZT levels (*p*-value = 0.000; MANOVA; *n* = 383). Shoot primordia formation was negatively correlated with increasing ZT levels in both genotypes, and M82 showed a significantly higher proportion of explants with prominent shoot primordia than MT (Figure 2a), where only 20% of explants were able to develop shoot primordia (*p*-value = 0.000; *n* = 383). In addition, the number of shoot primordia formed per explant at the end of the experiment was also strongly dependent on genotype (*p*-value = 0.003; *n* = 217; MANOVA), with M82 (9.3 ± 6.4 shoot primordia per explant; *n* = 188 explants) showing about a three-fold increase in the number of shoots formed compared to MT (3.4 ± 1.8; *n* = 29) (Figure 2b). Our results indicate that the best hormonal combination for callus growth in M82 and MT cotyledon explants was 1 mg/L of ZT and 0.1 mg/L of IAA (1.72 ± 0.73 cm^2^; *n* = 57; *p*-value = 0.002; LSD), whereas the best hormonal combination for higher induction of shoot formation was 1 mg/L of ZT and 0.5 mg/L of IAA (12.8 ± 9.7 shoot primordia per explant; *p*-value = 0.000; LSD; *n* = 73) (Figure 2c,d). In M82, primordia were observed early as darker filiform protrusions from the adaxial surface, with a higher frequency in a region close to the proximal incision where the dense callus develops (Figure 2e). Some spatial and temporal arrangement of these structures was observed at early time points (Figure 2f), although not all these primordia develop into functional shoots (Figure 2g,h), suggesting internal competition among primordia as a limiting factor necessary for subsequent shoot growth. Then, primordia develop near the distal incision and from the abaxial surface of the explant, as well as from inner regions of the lamina in those explants with homogeneous dense callus formation (Appendix A). In MT cotyledon explants, both primordia formation and functional shoot establishment was severely reduced in comparison to M82 (Figure 2a–d). In MT, these shoot primordia were more widely spaced, developed at a slower rate, and exhibited a lighter green color than in M82 (Appendix A). From our studies, the best hormonal combination for callus growth was 1 mg/L ZT and 0.1 mg/L IAA, and for higher induction of shoot formation, it was 1 mg/L ZT and 0.5 mg/L IAA.

### 2.2. Evaluation of the Regenerative Potential of Explants in Different Tomato Genotypes

To investigate whether the regenerative potential of the explants depends on the region of the cotyledon used, we cut 7-day-old cotyledons into three regions of similar size: apical, central, and basal (Appendix A). We found significant differences in explant area according to genotype (*p*-value = 0.000; MANOVA; *n* = 420), consistent with the larger size of cotyledons in the BGV016054, M82, and MM genotypes (Appendix A and Appendix A). The increase in explant area during the experiment was significantly dependent on genotype and region of the cotyledon used (*p*-value = 0.000; MANOVA; *n* = 409; Appendix A and Appendix A), with M82 showing the greatest increase in explant area, and the apical region of the explants showing the least growth in most genotypes. Callus formation was dependent on genotype and region of the cotyledon used (*p*-value = 0.000; MANOVA; *n* = 409; Figure 3a), with a weaker and positive effect of ZT levels (*p*-value = 0.018; Appendix A). Overall, callus formation reached ~40% in apical explants and ~70% in basal explants. Interestingly, 2 mg/L ZT positively increased callus formation only in apical explants and had no effect in central and basal explants. We also found that callus formation was significantly reduced (*p*-value = 0.000; LSD; *n* = 409) in explants smaller than 0.13 cm^2^ compared to intermediate size explants (Appendix A), suggesting that a minimum explant size is required for functional regeneration in tomatoes. BGV007927 and BGV016054 showed the lowest levels of callus formation, whereas M82 and MM showed the highest levels of callus formation from cotyledon explants, mainly from the central and basal regions of the cotyledon (Figure 3a).

We determined callus emergence visually and found that M82 showed significantly lower values (15.0 ± 2.0 days; *n* = 49) than the other genotypes (*p*-value = 0.000; LSD; *n* = 225), with the higher delay in callus emergence shown by BGV007927 (18.9 ± 1.4 days; *n* = 27). Interestingly, in M82, callus emergence occurred slightly earlier in apical than in basal explants (Appendix A). We used callus area as an estimate of callus growth during the experiment, and this trait was significantly dependent on genotype (*p*-value = 0.000; MANOVA; *n* = 229; Figure 3b). In addition, callus growth was also influenced by the region of the cotyledon used as an explant, with callus areas increasing from apical to basal explants in some genotypes (*p*-value = 0.006), such as M82 and BGV016054 (Figure 3b). Similarly, we also found a slightly positive and significant correlation between callus area and explant growth (r = 0.451; *p*-value = 0.000; Appendix A). Despite the high level of callus formation in MM, callus size was smaller than in MT (Figure 3b).

The time of de novo primordia formation was determined visually and found to be significantly influenced by genotype (*p*-value = 0.000; MANOVA; *n* = 135) but not by region (*p*-value = 0.207) or area of the explants (*p*-value = 0.060; Appendix A and Appendix A). Consistent with early callus formation in M82, this genotype showed the fastest initiation of primordia (17.3 ± 0.6 days; *n* = 40), whereas BGV007927 showed the greatest delay (27.5 ± 1.0 days; *n* = 13). These results suggest that in this later genotype, there is a delay of approximately 8.5 days between callus initiation and de novo primordia formation as compared to M82 (Appendix A). On the other hand, the proportion of explants forming primordia was dependent on genotype (*p*-value = 0.000; MANOVA; *n* = 247) and the region of the cotyledon used as an explant (*p*-value = 0.001; Figure 3c). MM showed the highest level of de novo primordia formation (~86%), whereas MT showed the lowest level of primordia formation (~24%). Consistently, basal explants showed the highest proportion of primordia formation among the genotypes studied (*p*-value = 0.007; LSD). Interestingly, the number of primordia formed was genotype dependent (*p*-value = 0.000; MANOVA; *n* = 134), with a slight effect of explant region (*p*-value = 0.087) but little contribution from ZT levels (*p*-value = 0.624). MT produced the lowest number of primordia per explant, 3.6 ± 2.4 (*n* = 11), while M82 and MM produced the highest number of primordia: 9.9 ± 5.6 (*n* = 38) and 10.1 ± 6.3 (*n* = 44) primordia per explant, respectively (Figure 3d). We found no significant correlation between explant size and number of primordia (r = −0.069; *p*-value = 0.427; Appendix A). We found that most of these primordia developed into shoot-like structures in less than one week (r = 0.978; *p*-value = 0.000). Consistent with these results, the number of shoot primordia was also genotype dependent (*p*-value = 0.000; MANOVA; *n* = 134), with a small effect of the explant region (*p*-value = 0.061) but no direct contribution of ZT levels (*p*-value = 0.838) (Appendix A), like that described above. In summary, callus formation was dependent on the explant region, ranging from ~40% in apical explants to ~70% in basal explants. M82 and MM were the two genotypes with high levels of callus and primordia formation. In this sense, M82 was the best genotype in primordia initiation, while MM presented the highest de novo primordia formation. For these parameters, cultivar MT showed the lowest values.

The presence of dense calluses was detected in most of the genotypes studied (BGV016054, BGV007910, M82, MM, and MT), while BGV007927 formed soft calluses from the beginning. The location of the callus induction showed some genotype-dependent differences (Figure 4). The location and structure of M82 and MT callus were mentioned above. MM formed calluses on both basal and apical sections of explants (Figure 4a,c). These calluses were small and light green in color. In addition, the explants of MM developed many round or oval callus regions of small size and whitish color within the lamina, in both the adaxial and abaxial surfaces (Figure 4b,c). BGV016054 and BGV007910, like M82, formed callus tissue mainly in the basal region near the cut end (Figure 4(d′,e′)). Small globular areas of whitish callus were detected on the adaxial side of both genotypes (Figure 4d,e). BGV007927 initially formed pale green or whitish callus tissue in the cut region, both apical and basal, and tends to transform almost the entire explant into callus tissue after some time in the culture medium. The first morphological structures observed in the callus tissue were round and protruded from the surface of the lamina and were probably the foci of formation of the new organs, which we will refer to as the globular phase (arrowheads in Figure 4). These globular structures continued to elongate and acquired an oval structure during the so-called filiform phase. Later, the filiform structures were transformed into small leaf primordium nodes, as indicated by the presence of newly developed glandular trichomes on their epidermis (asterisks in Figure 4). The color of the globular structures ranged from dark purple or almost black (M82; Figure 4f(f′)) to green (MT) or light green (BGV007910 and BGV016054; Figure 4d,e). These structures displayed variable color in MM and BGV007927 explants. The detailed localization of the globular structures and filiform structures of M82 (Figure 4g) and MT has been described above. MM, BGV007910, and BGV016054 structures were located near the cut ends and formed clusters. BGV007927 also initially formed primordia (single/pairs or small clusters) from the cut basal end of explants, but after about 1 month, when most explants became callus tissue, the primordia were randomly localized and associated with the callus site. Some of the shoot primordia became a functional meristem and produced stems available for rooting.

### 2.3. Cellular Features of De Novo Organ Formation in Tomato Cotyledon Explants

Due to the heterogenous response in callus and shoot morphology among the tomato genotypes studied, we decided to investigate the cellular characteristics that occur during de novo organogenesis using M82 basal cotyledon explants. As mentioned above, we observed that many leaf primordia were initiated at a certain distance from the basal incision and with a very regular spatial and temporal pattern (Figure 5a).

In order to localize cell cycle/DNA replications activity, we have used uridine analogue EdU, which is incorporated into nuclei exclusively during DNA replication and can be easily detected and quantified. SCRI Renaissance Stain 2200 (SR2200) was labeled cell wall (cellulose) and has been used to visualize and quantify cell structure, while DAPI has been used for visualization of cell nuclei and chromatin organization. Compared to neighboring soft callus tissues, cells in these incipient leaf primordia were characterized by smaller size and active DNA replication, as indicated by the strong staining of their nuclei with EdU (Figure 5b and Appendix A–C). Another interesting observation was that the isolated calli found on the cotyledon lamina consisted of many dividing cells on the surface and xylem cells in the inner tissues (Figure 5c and Appendix A). The callus tissue at the cotyledon margin consisted of small isodiametric cells with high EdU staining of their nuclei (Figure 5d), indicating their meristematic potential. A cell division gradient from the cotyledon margin to the inner lamina was clearly observed (Figure 5e). On the other hand, the soft callus tissue was characterized by large cells with brightly stained cell walls (Figure 5f), which occasionally divided (Figure 5g). Taken together, our results indicate differential tissue-specific responses in tomato explants to the addition of external hormones that merit further investigation.

## 3. Discussion

In this work, we optimized the protocol for hormone-induced shoot regeneration in tomato cotyledon explants using two widely used tomato genotypes, M82 and MT, and four other cultivars with highly divergent genetic backgrounds (see Section 4). We found that tomato cotyledon explants, regardless of their genotype and their regenerative responses, increased in size during the experiment, indicating that they contain sufficient resources (hormones, photosynthates, etc.) to maintain autonomous growth of the explants for several weeks, in contrast to what occurs in *Arabidopsis thaliana* cotyledons, whose postembryonic growth depends mainly on cell expansion [18].

Previous studies in *A. thaliana* have shown that treatment with the exogenous CKs 6-benzylaminopurine (BAP) or ZT induces endogenous IAA biosynthesis and increases steady-state auxin levels in young shoots and roots [19]. Addition of CKs can also affect endogenous auxin distribution by regulating the PIN-FORMED (PIN) auxin efflux transporters [20]. Our results on callus and adventitious shoot formation in tomato cotyledon explants in response to exogenous treatment with ZT and IAA also suggest a complex regulation of endogenous auxin and CK levels. Interestingly, exogenous auxin significantly enhanced the effect of ZT on callus formation and de novo shoot formation, whereas higher levels of ZT negatively affected de novo shoot formation. Indeed, in the absence of IAA, high levels of ZT (3 mg/L) significantly enhanced the callus formation response in the MT background but negatively regulated callus growth in this genotype, as this is highly dependent on the endogenous auxin-to-CK ratio, as described in other *Solanaceae* species [21]. At low ZT levels (1 mg/L), adventitious shoot production in M82 was dependent on increasing exogenous IAA levels. However, the higher regenerative response of M82 over MT in terms of de novo shoot induction could be because the MT background contains mutations related to gibberellin (GA) signaling and brassinosteroid biosynthesis [5], which could directly affect shoot apical meristem (SAM) activity, as has recently been shown for the positive role of GA in SAM growth in *A. thaliana* [22].

In a previous report, the addition of 0.05 or 0.1 mg/L IAA to ZT-containing growth medium reduced the recovery time of regenerated M82 plantlets by 6 weeks [13]. In another work, plant regeneration by indirect organogenesis was developed for 7-day-old cotyledon explants of four tomato cultivars (Rio Grande, Roma, hybrid 17905 and M82) in Murashige and Skoog (MS) medium supplemented with different combinations and concentrations of plant growth regulators (PGRs) [16]. Regenerating calli resulted in the formation of multiple shoots after 3 weeks on medium containing 3 mg/L BAP and 0.1 mg/L IAA. Furthermore, Sandhya et al. (2012) reported that 2.0 mg/L ZT in combination with 0.1 mg/L IAA produced the highest number of shoots from the cotyledon explants of two different tomato cultivars, Arka Vikas and Pusa Early Dwarf [23]. Shorter explant-to-plantlet regeneration time is always desirable, as it reduces labor and resource costs and allows high-throughput approaches for early recovery and evaluation of transgenic lines. These and other results [24] suggest a genotype × environment (i.e., PGRs) interaction in the regenerative response of tomato cotyledon explants. We speculate that the addition of IAA to the growth medium facilitates auxin and CK crosstalk within the explants, which is then required for the acquisition of competence to establish de novo shoot apical meristems by specific developmental regulators [25]. In our work, we avoided the use of 2,4-dichlorophenoxyacetic acid, which is routinely used in indirect somatic embryogenesis but is known to cause morphological abnormalities in different species because of physiological disorders or somaclonal variation [26].

We investigated the effect of the region of the cotyledon (basal, central, and distal) used as explants on callus formation and de novo shoot induction in six tomato genotypes. In all cases, callus growth was observed at both cut ends (proximal and distal) of each explant, shortly after excision, and with greater growth of callus tissue in the proximal region of the explants regardless of the region of the cotyledon used. Callus growth in the proximal region of the explants is likely dependent on endogenous auxin gradients that are dynamically established by the endogenous PIN-mediated polar auxin transport system, as previously demonstrated in tomato hypocotyl explants [27,28]. Callus formation is achieved in all six genotypes at different rates with a region × genotype dependence: in general, the basal region of the cotyledon is more responsive than the apical one, and the studied heirloom tomato cultivars are less responsive than the commercial ones. The minimum size of explants that produced effective regeneration under our conditions was approximately 0.20 cm^2^, and the average time for callus establishment ranged from 15 to 19 days. De novo shoot primordia formation was strongly genotype dependent, with MM showing the highest number of fully developed shoot primordia. For most genotypes, explants from the basal region of the cotyledon showed the highest number of developed shoot primordia. In addition to MT, two heirloom varieties from the Varitome collection [7] that are from México, BGV007910 and BGV007927, showed a low proportion of adventitious shoots formed. These results suggest interesting genotype-dependent regenerative responses in the Varitome collection that deserve further investigation.

Previously, Chaudry et al. (2010) [29] investigated the regenerative capacity of hypocotyl and leaf discs from MM explants, using a combination of ZT and IAA, which induced de novo shoot formation from hypocotyl explants but not from leaf discs. Lee et al. (2020) found that the optimal culture conditions for de novo shoot formation from MT explants were MS mediums containing 1 mg/L ZT, 0.1 mg/L IAA, and 3% sucrose. They also found non-significant differences in de novo shoot formation with respect to explant age up to 10 days after germination. Consistent with our results for several tomato genotypes (see above), de novo shoot formation in MT was much higher in the basal cotyledon explants than in the central and apical explants, with a greater positive effect on whether the abaxial side of the explant touched the medium [16]. Interestingly, the region of the hypocotyl closest to the cotyledons produced the highest number of de novo shoots, which correlated with higher expression of the shoot identity genes WUS and STM in this region [16]. In a recent investigation, Sundhya et al. (2022) [30], using cotyledon explants from two Indian tomato cultivars, demonstrated that ZT was the most effective CK for shoot formation compared to BAP or thidiazuron. Taken together, our results indicate that the optimal conditions for de novo shoot formation in tomatoes are the use of the basal segment of young cotyledons (7-day-old) placed in culture medium (MS, 1 mg/L ZT, 0.1 mg/L IAA, 2% sucrose) on their abaxial surface.

The callus tissues formed in tomato explants in response to treatment with ZT and IAA were heterogeneous in nature. Macroscopically, we distinguished between soft or friable calli, usually found at the edges of the explants and near the cut region, and dense calli, the latter composed of small, isodiametric, and actively dividing cells. Soft calli are composed of multicellular structures resembling trichomes. These structures grow polarly outward from the explant surface by limited cell division and mainly by cell expansion. This soft callus tissue is disorganized and rarely produces adventitious shoots. In some of the genotypes studied, such as BGV0016054 or MT, the soft callus tissue prevailed.

Dense calli are normally found in the cotyledon margin and within the cotyledon lamina. Interestingly, a decreasing cell division gradient in palisade mesophyll cells is observed from the cotyledon margin, as evidenced by the EdU/DAPI nuclear ratio (this work). These results suggest that cells at the cotyledon margin retain their meristematic potential. Some mesophyll cells adjacent to this tissue also replicate DNA but rarely divide, resulting in cell growth by endoreduplication. In *A. thaliana*, a complex regulation of transcription factors contributes to leaf margin growth, although the molecular mechanism remains to be elucidated [30,31]. We speculate that exogenous ZT could activate the marginal meristem of tomato cotyledons, probably through the WUSCHEL-RELATED HOMEOBOX1 (WOX1) and WOX3 transcription factors, which in *A. thaliana* are known to positively regulate the expression of *YUCCA1* (*YUC1*) and *YUC4* to synthesize auxin in the margin and thus control lamina outgrowth [32]. Dense calli found on the cotyledon lamina are highly structured, with many dividing cells in the epidermis and palisade mesophyll resembling immature shoot meristems, while new xylem cells form in the spongy mesophyll just below, presumably by hormone-mediated transdifferentiation of these cells [33]. These results suggest that within dense calli, some compartmentalization of CK and auxin responses is expected, leading to cell cycle activation in the epidermal/palisade mesophyll cells and near the margin, and increasing gradients of cell differentiation from the margin to the cotyledon lamina as well as from the epidermal/palisade mesophyll cells into the spongy mesophyll cells in response to newly formed adaxial-abaxial and mediolateral auxin gradients. The availability of *DR5:mScarleti-NLS* and *TCSn:mNeonGreen-NLS* tomato plants [34] will allow precise observation of endogenous auxin and CK response gradients during callus formation and de novo shoot induction, which will help clarify our histological observations.

A competitive balance between callus formation and shoot regeneration has been proposed. In *A. thaliana* explants, callus tissue resembles a root primordium with active auxin signaling [35]. However, shoot regeneration is dependent on CK signaling as well as the repressing of root identity [36]. Multiple shoots were initiated at both the adaxial and abaxial surfaces of the dense callus tissue, mainly near the basal region of the cotyledon explants but not at the cutting edge where soft callus tissue is formed. In analogy to the regeneration model for tomato hypocotyl explants [27,28,37], wound-induced local auxin biosynthesis and endogenous polar auxin transport pathways might create an auxin gradient that is maximal at the cut end of the explants, leading to soft callus production. In the presence of ZT, a high CK-to-auxin ratio is established in the region distal to the cut end of the explants, which may be sufficient to induce dense callus formation with SAM identity foci. A similar scenario could explain the formation of shoot primordia within the globular callus on the cotyledon lamina, where cells near the surface actively divide in response to the CK treatment and establish new auxin sinks in the inner mesophyll, causing transdifferentiation of spongy mesophyll cells into xylem bundles that establish new polar auxin transport routes from the dividing cells of the callus to the internal vasculature, generating a minimum of auxin in some of the dividing cells at the adaxial side that could contribute to the specification of new SAMs in this region.

Comparison of our study with [38] demonstrated that our hormone combination (ZT/IAA) can provide a greater number of shoots in the MM cultivar. Moreover, our conditions also showed a greater number of the shoots per explants as previously reported [39].

In conclusion, our study extends the knowledge of the regenerative capacity of cotyledon explants through the analysis of three tomato cultivars widely used in research and three ancestral tomato lines from the Varitome collection. In this sense, we have determined the optimal concentrations of 1 mg/L ZT + 0.5 mg/L IAA, as well as the regions of the cotyledon that showed the highest rate of regeneration. Finally, we performed a histological study of de novo shoot formation, which adds to the knowledge of this process. Our presented method has the advantage of being carried out in 6-well plates, which allows high throughput escalation, as well as rapid regeneration, and can be applied to different tomato varieties. These advantages are very useful for plant transformation and gene editing experiments.

## 4. Materials and Methods

### 4.1. Plant Materials and Growth Conditions

*Solanum lycopersicum* L. seeds were germinated and cultivated as previously reported [40]. In our experiments, we used three well-known tomato cultivars, M82, MM, and MT, as well as three other lines from the Varitome collection: BGV007910, BGV007927, and BGV016054 [8] (Table 1). For E1, E2, and E3 experiments, 7-day-old cotyledons were cut in half (0.51 ± 0.18 cm^2^) and incubated in 6-well plates with MS basal salt medium containing 2% sucrose and 0.8% agar supplemented with ZT (1, 2 or 3 mg/L) or IAA (0, 0.1 and 0.5 mg/L), with the abaxial side in contact with the agar surface. For the E4 experiment, 7-day-old cotyledons were cut into three regions of similar size, apical, central, and basal (0.27 ± 0.17 cm^2^), and incubated in an MS medium containing 2% sucrose and 0.8% agar supplemented with 1 or 2 mg/L ZT and 0.5 mg/L IAA. Cultivation conditions were 16 h LED light/8 h dark period and 23–24 °C for all experiments. Experiments were performed in three biological replications with a minimum of 15 cotyledons per genotype and treatment (E1, E2 and E3; *n* = 585 explants) or with 12 cotyledons per genotype and treatment (E4; *n* = 432 explants). Plates were incubated for 6 weeks, with explants periodically transferred every 2 weeks to freshly prepared media containing a similar combination and concentration of growth stimulants. Plates were scanned weekly, and the presence of calluses and shoot primordia was visually assessed using a stereomicroscope. Explant and callus areas were quantified from scanned images using ImageJ [41] after freehand line drawing. Callus/primordia/shoot emergence was defined as the first day of callus/primordia/shoot observation after daily examination of each explant under a stereomicroscope. Shoots were distinguished from primordia by the presence of glandular trichomes.

### 4.2. Statistical Analyses

Statistical analyses of the data and descriptors (mean, SEM, maximum and minimum, and correlation values) were estimated using StatGraphics Centurion XVI version 16.1.03 (StatPoint Technologies, Warrenton, VA, USA). Outliers were identified and were excluded for subsequent analyses, as described elsewhere [42]. To compare data for a given variable, we performed multiple testing analyses using the multivariate ANOVA, F-test, or Fisher’s least significant difference post hoc test, as indicated. Significant differences were defined as a significance level of 1% (*p*-value < 0.01), unless otherwise indicated. To determine the correlation between the different parameters, multiple correlation tests were performed.

### 4.3. Microscopic Observation and EdU Staining

To visualize the origin of the cell cycle progression, 7- to 10-day-old cotyledon explants grown on regeneration medium for 3–4 days were transferred to liquid medium with the same composition, and EdU was added for the indicated pulse labeling time. Explants were fixed, and EdU was detected as previously described [43]. Samples were stained with DAPI and mounted on slides with double spacers (300 μm thick). To characterize tissue structure, cotyledon explants with de novo shoot primordia were fixed in 4% formaldehyde solution for 30 min, cleaned as described to increase tissue transparency, and partially digested for cell wall and membrane [43,44]. Explants were incubated with the EdU detection solution for 30 min and then incubated in MTSB buffer (pH 8.1) supplemented with 1:1000 SCRI Renaissance 2200 (SR2200; Renaissance Chemicals Ltd., Selby, UK) and mounted on slides with double spacers. Images were captured using a Leica STELLARIS STED microscope (Leica Microsystems, Wetzlar, Germany). For SR2200 and DAPI: ex. 405; for Alexa 488 (EdU): ex: 488. Image analyses were performed as described [44].

## Figures and Tables

**Figure 1 plants-12-02942-f001:**
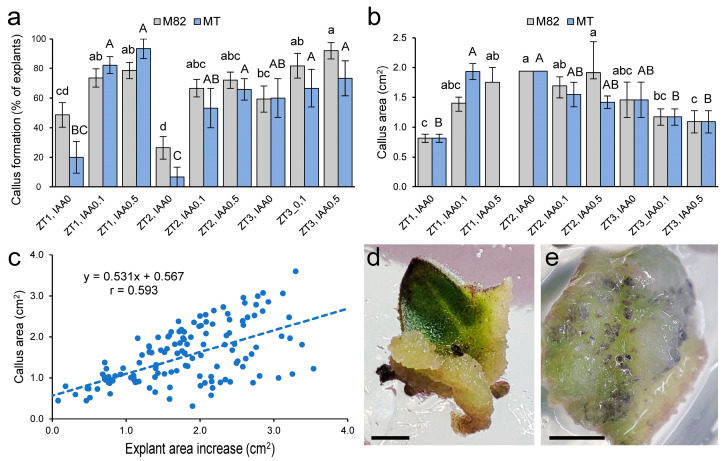
Hormone-induced callus formation in M82 and MT tomato cultivars. (**a**,**b**) Callus formation (**a**) and callus area (**b**) in response to ZT and IAA treatment. Bars indicate mean and standard error of the mean (SEM). Letters indicate significant differences between treatments for a given genotype (*p*-value < 0.01; Fisher’s least significant difference [LSD]). Raw data is shown in Appendix A. (**c**) Correlation between explant and callus area. (**d**,**e**) Representative images of callus formation in distal (**d**) and proximal (**e**) cotyledon explants at 21 days. Scale bars: 3 mm.

**Figure 2 plants-12-02942-f002:**
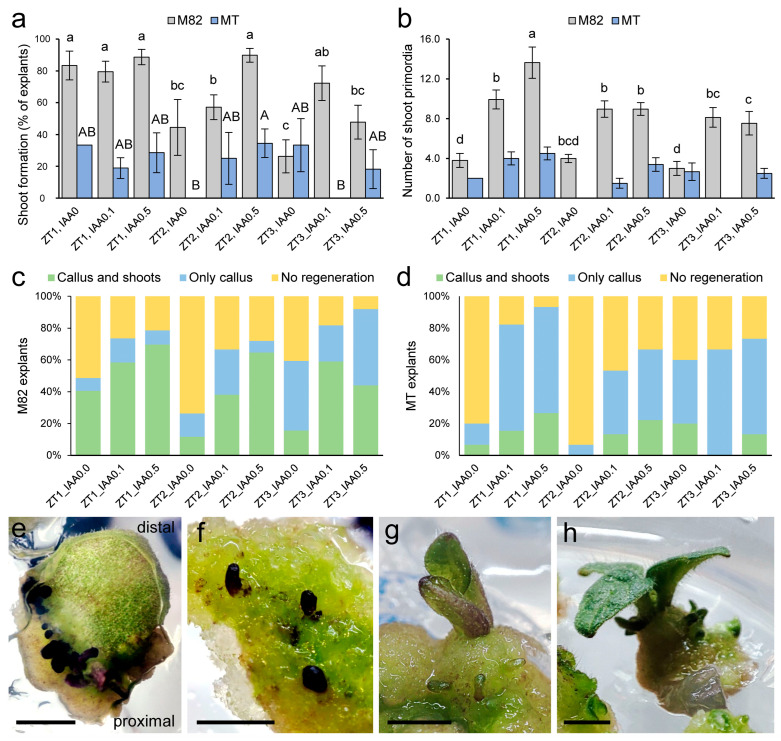
Hormone-induced shoot formation in M82 and MT tomato cultivars. (**a**,**b**) Shoot formation (**a**) and number of shoot primordia (**b**) in response to ZT and IAA treatment. Bars indicate mean and SEM. Letters indicate significant differences between treatments for a given genotype (*p*-value < 0.05; LSD). Raw data is shown in Appendix A. (**c**,**d**) Regenerative response of M82 (**c**) and MT (**d**) explants in response to ZT and IAA treatment. (**e**–**h**) Representative images of a time series for shoot formation in M82 cotyledon explants at 21 (**e**,**f**), 25 (**g**), and 55 (**h**) days. Scale bars: 3 mm.

**Figure 3 plants-12-02942-f003:**
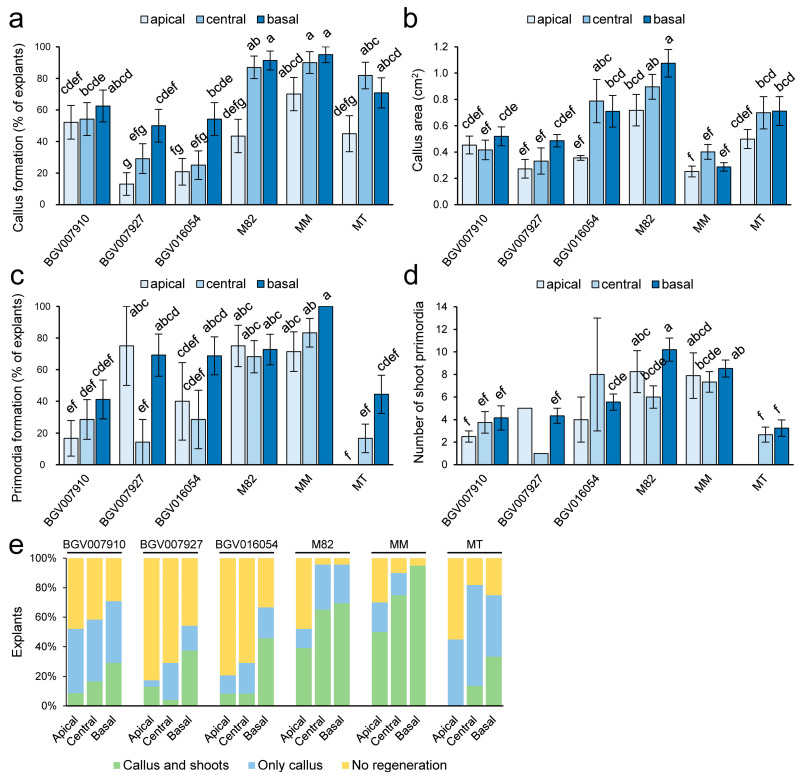
Hormone-induced callus and shoot formation is genotype and region dependent. (**a**,**b**) Callus formation (**a**) and callus area (**b**) in the studied genotypes from different regions of the co- tyledon. (**c**,**d**) Shoot formation (**c**) and number of shoot primordia (**d**) in the studied genotypes from different regions of the cotyledon. Bars indicate mean and SEM. Letters indicate significant differences between samples (*p*-value < 0.01 (**a**,**c**) or *p*-value < 0.05 (**b**,**d**); LSD). Raw data are shown in Appendix A. (**e**) Regenerative response of the studied genotypes with respect to the different regions of the cotyledon.

**Figure 4 plants-12-02942-f004:**
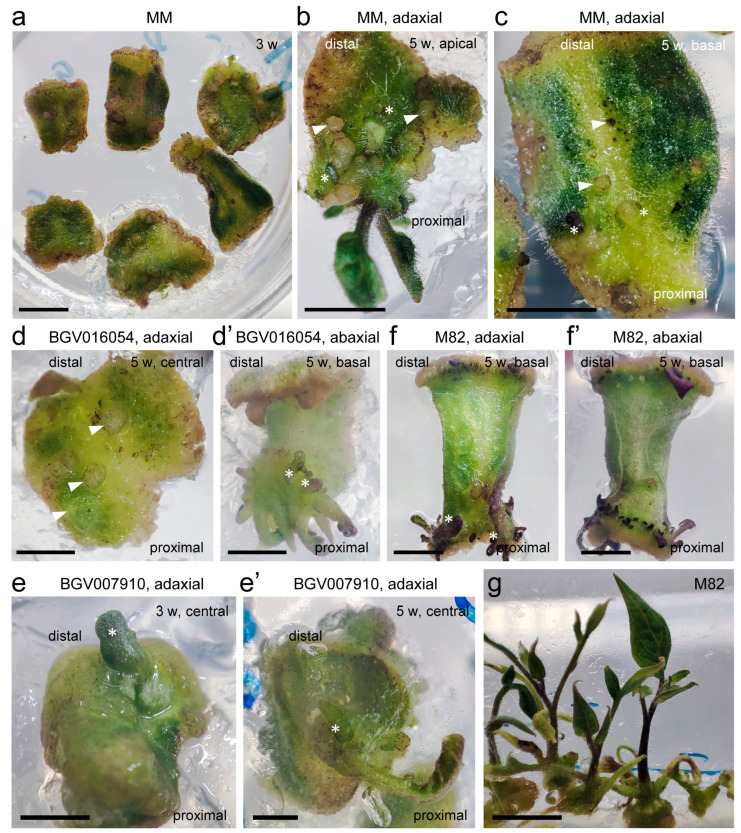
De novo organ formation in excised cotyledon explants of some of the genotypes studied. (**a**–**c**) MM [representative examples of apical (**b**) and basal (**c**) explants are shown] at 32, 39, and 24 days, respectively; (**d**,**d′**) BGV016054 at 19 days, (**e**,**e′**) BGV007910 at 24 and 19 days, respectively, and (**f**,**f′**) M82 at 21 days. (**d**,**f**): adaxial side; (**d′**,**f′**): abaxial side; arrowheads point to the globular structures within the callus tissue, and asterisks indicate some of the developing leaf primordia. (**g**) M82 explants with elongated shoots before rooting at 80 days. Scale bars: 3 mm.

**Figure 5 plants-12-02942-f005:**
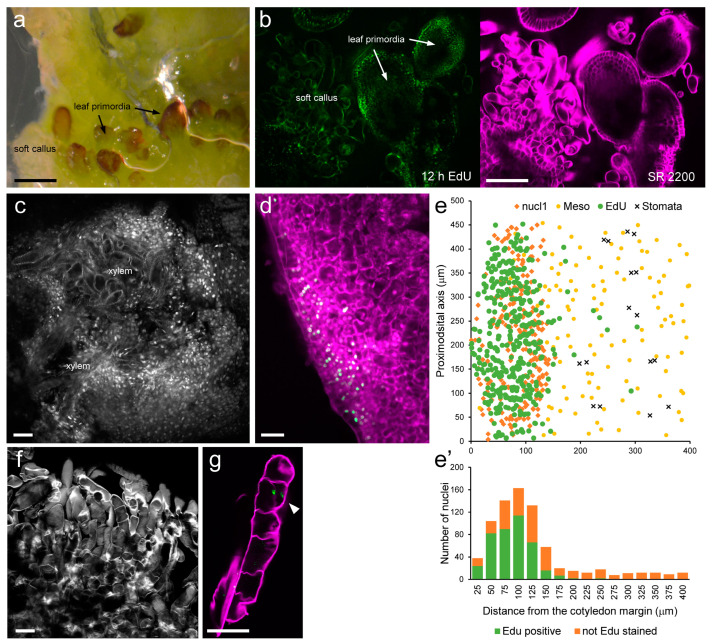
Cellular features of de novo organ formation in tomato cotyledon explants. (**a**) Detailed observation of hormone-induced shoots in M82 cotyledon explants at 3 weeks, showing soft callus tissue and incipient leaf primordia. (**b**) Microscopic observation of the explant shown in (**a**), incubated for 12 h with 5-ethynyl-2**′**-deoxyuridine (EdU; green) and stained for cell walls with SR2200 (magenta). (**c**) Detailed observation of a globular callus tissue from (**a**) stained with DAPI, where xylem cells are clearly observed. (**d**) Details of the cotyledon margin in M82 cotyledon explants at 4 days on regeneration medium, incubated for 4 h with EdU (green) and stained with DAPI (magenta). (**e**) Nuclear coordinates from (**d**,**e′**) histogram of the percentage of EdU-positive cells. Nuclei and EDU positive nuclei were detected and plotted in Cartesian coordinates. (**f**,**g**) Microscopic observation of soft callus tissue (**g**) and details of a dividing cell (EdU, green, white arrow; SR2200, magenta). Scale bars: 2 mm (**a**) and 40 µm (**b**–**g**).

**Table 1 plants-12-02942-t001:** Tomato genotypes studied in this work.

**Accession**	**Organism**		**Name**	**Genotype ^1^**
LA3475	*S. lycopersicum* var. *lycopersicum*		M82	*sp*; *u*; *obv*; *I*; *Ve*
LA3911	*S. lycopersicum* var. *lycopersicum*		Micro-Tom	*d*; *sp*; *ej-2^w^*; *u*; *I*; *Sm*
LA2706	*S. lycopersicum* var. *lycopersicum*		Moneymaker	*sp^+^*; *ej-2^w^*; *u*; *obv^+^*
**Accession**	**Organism**	**Country**	**Site**	**Latitude, Longitude**
BGV007910	*S. lycopersicum* var. *cerasiforme*	México	Palo de Arco; Ciudad Valles; San Luis de Potosí	21.91, 99.16
BGV007927	*S. lycopersicum* var. *cerasiforme*	México	El Vergel: Culiacán. Sinaloa	24.73, 107.79
BGV016054	*S. lycopersicum* var. *cerasiforme*	Perú	Rumizapa	6.45, 76.47

**^1^***d*: dwarf; *ej-2^w^*: enhancer of jointless-2^weak^; *I*: Immunity to Fusarium wilt; *obv*: obscuravenosa; *obv^+^*: obscuravenosa^clear vein^; *sp*: self-pruning; *sp^+^*: self-pruning^+^: self-pruning^wild-type allele^; *Sm*: Stemphyllium resistance; *u*: uniform ripening; *Ve*: Verticillium resistance.

## Data Availability

All data generated or analyzed during this study are provided in this published article and its Appendix A, or they will be provided upon a reasonable request.

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
