# Peer review of "Optimization of Callus Induction and Shoot Regeneration from Tomato Cotyledon Explants"

_plants, 2023, doi:10.3390/plants12162942_

Round 1

Reviewer 1 Report

I suggest making minor corrections and additions - details have been marked in the attached file.

Author Response

We thank this reviewer for his/her useful suggestions. We provide a detailed point-by-point response to his/her comments in the attached PDF.

Reviewer 2 Report

The manuscript, as a whole, is devoted to an important problem of plant regeneration from different explants for various genotypes of tomato that is necessary for tomato genetic improvement.  The content of the manuscript is clear, relevant for the field and presented in a well-structured manner.

At the same time, it is necessary to note that  a number of regeneration protocols have already been published for  the cultivars studied in this work,  for  example for cv. Moneymaker (a model tomato plant): in the work of Chaudhry  Zubeda, Abbas Sidra, Yasmin Azra et al. (2010). Tissue culture studies in tomato (Lycopersicon Esculentum) var. Moneymaker. Pakistan Journal of Botany. 42; or in the work NAVEEDA ANJUM et al., (2014) In Vitro Multiple Shoot, Root and Callus Induction from Different Explants in Tomato (Lycopersicon Esculentum Mill). Journal of Biology and Medical Sciences Vol. 2, 2014, and so on.

Thus, the authors have to explain what results are scientifically sound and novelty in their study?

It is necessary to refine and strengthen the Discussion section. In particular, the comparison of experimental data should be made with already published data for tomato varieties, in particular for the cv. Moneymaker, and not for Arabidopsis.

Secondly, it is known, that genetic uniformity is one of the most essential requisites in in vitro culture and regeneration of plant species. Authors declaimed, that their “method is suitable for obtaining many seedlings of different tomato genotypes”. Hence, it could be necessary to present the results of molecular analyses of direct and indirect regenerants, confirming the occurrence of true-to-type plantlets. For example data of genetic fidelity assessment using ISSR (Inter Simple Sequence Repeats) markers.

Additionally, the cited references of manuscript need to cover the topic of tomatoes to a greater extent. At the moment, out of 41 items, only 20 articles on tomatoes, and only 27 of them are the most recent publications (within the last 5 years).

Author Response

(The authors gave the same response as above.)

Reviewer 3 Report

Dear authors,

Hope you are doing well.

I read your manuscript about callus induction and shoot regeneration from cotyledons of tomato in vitro plants. Globally, I think that this research is very interesting because explain and confirm some basic aspects of in vitro culture. Additionally, provide useful information about the propagation of tomato in vitro plants s for genomic applications.

I believe that the article is well written. However, I have some minor concerns about figures (and tables) and results description. These comments are aimed at a better comprenhension and reading.

Figures, legends and tables:

- Supplementary tables. Please, include a description of abbreviations at the final of the table. For example, T0w, T1w,...

- Figure 1b and 2b. Some bars are missing, what is the meaning of the absence? (lost data, problems during experiment, not callus formation,...). Please, explain this and include it in the text if it is appropiated.

- Several figures. I suggest to include the evaluation time for each experiment in the legend.

Description of results:

- I suggest to include a brief sentence about the aim of the experiment (line 84).

- Because your results have many numerical values, they can be difficult to read and understand for readers. Therefore, I recommend including a shor summary paragraph with the most important results.

Other minor concerns:

- Table 1. "var" for "var."

- Line 53. Please, explain that you are using only S. lycopersicum var. cerasiforme in your experiment.

- Line 78. You mention "position of the explants". What is the meaning of that? From my point of view, it sounds like the location of leaves in the stem of the plant, but I think you mean basal, apical,...

- Line 85 and 120. Please, homogenize: USE "three-fold" or "three-fold".

Congratulations for your work!

Best regards,

Author Response

(The authors gave the same response as above.)

Reviewer 4 Report

The following MS "Optimization of callus induction and shoot regeneration from tomato cotyledon explants" by Yaroshko, described the optimization methods for callus induction and shoot regeneration from tomato cotyledon explants from tomato cultivars namely, M82 and Micro-Tom. The work was designed and executed well. The flow and English of the MS is fine.

Comments:

1. Why author specifically choose Zeatin instead BAP for their study. Is there any specific reason behind this?

2. A conclusion with clear outcomes and future direction should be added at the end of discussion.

3. Line no. 308, on what combinations auxin used for shoot regeneration from cotyledon explants.

4. What is the major significance of the current work differed with Sandhya et al., (2012) in terms of shoot regeneration and callus induction efficiency.

5. Check the typo errors. ex. Line no. 379. - ot to of

6. Why don't author have not studied the Agrobacterium transformation efficiency since it will be final application target?

Author Response

(The authors gave the same response as above.)

Round 2

Reviewer 2 Report

"It should be noted that the authors responded to some of our comments, and methodically the work was done in good faith. However, the absence of molecular analysis data, and the general results do not have sufficient relevance and novelty, by our minds."

Author Response

"It should be noted that the authors responded to some of our comments, and methodically the work was done in good faith. However, the absence of molecular analysis data, and the general results do not have sufficient relevance and novelty, by our minds."

Thanks to the reviewer for his comment. In the article we have tried to systematically explore which is the best combination of hormones for de novo regeneration in representative tomato species, such as M82, Money Maker and Micro-Tom, as well as in older species obtained from the Varitome collection. Although it is true that molecular studies have not been carried out, we have addressed regeneration using cell biology techniques, which have not been applied to this type of study.